# Contribution of Different Brain Disorders and Multimorbidity to Delirium Superimposed Dementia (DSD)

**DOI:** 10.3390/geriatrics8030064

**Published:** 2023-06-02

**Authors:** Tilman Wetterling, Klaus Junghanns

**Affiliations:** 1Department of Psychiatry, Vivantes Klinikum Kaulsdorf, 12621 Berlin, Germany; 2Campus Lübeck, University Hospital of Schleswig-Holstein, 23538 Lübeck, Germany; klaus.Junghanns@uksh.de

**Keywords:** dementia, delirium, multimorbidity, brain infarction, white matter hyperintensities

## Abstract

Delirium, an acute neuropsychiatric disorder characterized by a disturbance of attention and awareness, is often superimposed on dementia with its progressive cognitive decline. Despite the high frequency and clinical relevance of this condition, often called delirium-superimposed dementia (DSD), little is known about possible triggers. In this study using the GePsy-B databank, we investigated the impact of the underlying brain disorder and multimorbidity (MM) on DSD. MM was measured by CIRS and the number of ICD-10 diagnoses. Dementia was diagnosed by CDR, and delirium by DSM IV TR criteria. A total of 218 patients were diagnosed with DSD and these were compared to 105 patients with only dementia, 46 with only delirium, and 197 patients suffering from other psychiatric diseases, mainly depression. No significant differences between groups were found concerning CIRS scores. Based on CT scans, DSD cases were grouped into those with cerebral atrophy only (probably pure neurodegenerative), with brain infarction, or with white matter hyperintensities (WMH), but no between-group differences regarding the MM indices could be found. Regression analysis only revealed age and dementia stage as influencing factors. Conclusion: Our results suggest that neither MM nor morphologic changes in the brain are predisposing factors for DSD.

## 1. Introduction

In the hospital and in nursing homes, the care of demented individuals is often complicated by delirium [1]. The prevalence of delirium superimposed on dementia (DSD) ranges from 22% to 89% of hospitalized and community populations with dementia [2,3]. There is a close interaction between delirium and dementia since dementia is a risk factor for delirium, and delirium is a known risk factor for newly developed dementia or worsening of dementia [4,5,6,7,8]. However, there is considerable overlap in the psychopathological symptomatology of both neurocognitive disorders complicating the comparison of published data on demented individuals [4,5,9].

The characteristic features of delirium are acute onset and fluctuating symptoms, including inattention, level of consciousness, and cognitive disturbances [10,11,12]. Its etiology remains poorly understood. Delirium may occur in response to a variety of noxious insults, including medications, infections, and surgery [13,14]. Individuals with preexisting diminished cognitive status may be most vulnerable to developing delirium.

Dementia is mostly caused by slowly progressive neurodegenerative disorders, suchas Alzheimer’s disease (AD). Most studies on DSD primarily investigated AD patients. AD is characterized by impaired memory and loss of ability to function independently. However, dementia and delirium is also frequently induced by vascular disorders [15,16]. The rate of DSD seems to be higher in vascular dementia than in AD [17].

However, there are different potential pathogenic pathways for vascular dementia [18]; among them, cerebral small vessel disease (cSVD) is the most frequent [19]. Evidence from neuroimaging studies shows that delirium may mainly be associated with white matter hyperintensities (WMH) [20,21]. WMH are supposed to be comprehensive expressions of the pathological changes of cSVD [19,22].

Delirium is considered to have a complex multifactorial etiology. Many predisposing and precipitating factors are known [13,14]. An important predisposing factor may be the underlying brain disorder, which in vivo can best be investigated by neuroimaging [16,21]. Despite the high frequency and clinical relevance of delirium-superimposed dementia, little is known about possible triggers. Since multimorbidity (MM) presents similar to delirium, a clear age dependence [23], the question raises whether MM may contribute to the development of DSD.

This study will assess the hypotheses: first, different kinds of underlying brain damage, such as neurodegenerative diseases, infarction, or WMH contribute to the severity of DSD. Second, DSD patients suffer from more pronounced MM than patients with dementia alone or delirium alone. 

## 2. Materials and Methods

The complete data set of the GePsy-B study (medical records of 941 neuropsychiatric inpatients >65 years) was used for this study, described in detail in [24]. The GePsy-B study was performed in a hospital which provides acute psychiatric service for about 250,000 inhabitants of an eastern district in Berlin, Germany. Two wards are specialized for the care of neuropsychiatric patients. Only first admissions during the study period were considered. 

### 2.1. Subjects

A total of 274 subjects had to be excluded because of incomplete data. This incompleteness was mostly caused by difficulties in performing CT scans as the patients showed agitation or other behavior disturbances. In 56 cases, CT scans revealed cerebral tumors or chronic subdural hematoma, etc. (Figure 1). A total of 45 were not included because of recent surgical operations, probable diagnoses of Parkinson’s disease, Lewy body disease, or withdrawal from psychotropic substances as a possible cause of DSD.

Thus, our sample consists of 566 patients, mean age: 79.5 + 7.7 years. On average, the 382 female patients were significantly older than the 184 male patients (80.1 + 7.8 vs. 78.1 + 7.4 years) (U-test *p* = 0.004).

### 2.2. Measures

Delirium and dementia, according to DSM IV TR criteria [25], were diagnosed by a senior psychiatrist. Delirium was assessed at admission, then daily during the first week by using the Confusion Assessment Method (CAM-S) [26] in the validated German translation [27] and the Delirium Rating Scale-Revised-98 (DRS-R-98) [28] 2001). The maximum values were calculated. The stage of dementia was assessed by the Clinical Dementia Rating (CDR) [29].

Multimorbidity was measured retrospectively by the Cumulative Illness Rating Scale (CIRS) [30] as described in detail by Salvi et al., 2008 [31]. To isolate any effect of dementia or delirium, the neuropsychiatric item of the CIRS was eliminated, leaving 13 systems to comprise a total score, referred to henceforth as CIRS-13. The count of ICD-10 diagnoses was a simple and universally available comorbidity index. Furthermore, the nutritional status was evaluated by using the Mini Nutritional Assessment short-form (MNA-SF) [32].

As outcome variables, the days of inpatient treatment, the mortality rate, the Charlson Comorbidity Index (CCI) [33], and the Clinical Global Impressions (CGI) Scale [34] at discharge were used. The CCI was primarily developed to prospectively assess mortality risks due to comorbid conditions.

Since MRI scans were conducted only in a small portion of the patients, the findings of cerebral CT scans were used. All other data, such as routine laboratory values, length of stay, etc. were gathered from the anonymized medical records. The cut-off values of laboratory parameters are given in Tables 4 and 5.

### 2.3. Statistical Analysis

The statistical calculations were conducted with the IBM-SPSS program, version 27.0G. One-step ANOVA was computed. The homogeneity of variances was tested by Levene’s test. In the case of inhomogeneity, nonparametric tests, such as the Mann–Whitney U-test or the Kruskal–Wallis H-test, were performed. Correlations were calculated as Pearson’s r. The level of significance was given in the text as * *p* < 0.05, ** *p* < 0.01, and *** *p* < 0.001.

## 3. Results

A three steps analysis was performed. In the first step, we calculated the data of all patients. The degree of dementia as measured by CDR showed a clear age dependence: no dementia (CDR < 1) 243 cases, mean age 76.8 + 7.6 years; mild dementia (CDR = 1) 103, mean age 81.2 + 7.5 years; moderate dementia (CDR = 2) 120 mean age: 81.5 + 6.3 years; and severe dementia (CDR = 3) 100, mean age: 81.8 + 7.8 years (H-test 79.9 ***).

The number of typical delirium symptoms as assessed by CAM-S or DSR-98 scores showed a significant age dependence (r = 0.195 *** or resp. 0.201 ***), as did the numbers of somatic ICD-10 diagnoses (r = 0.175 **). The CAM-S and DSR-98 scores correlated with the number of ICD-10 diagnoses (r = 0.099 * or resp. 0.119 **) and strongly with the severity of dementia measured by CDR (r = 0.530 *** or resp. 0.581 ***).

In 160 subjects, CT-findings consisted of atrophy only (age: 77.2 + 7.0 years); in 82 subjects (mostly non-territorial) infarcts could be seen (age: 81.2 + 7.6 years) and 324 patients had WMH (age: 80.2 + 7.8 years) (ANOVA df 2 F10.8 ***).

In the second step, the sample was divided into four groups according to their current psychopathological symptomatology: dementia alone, delirium alone, DSD, and other diagnoses (i.e., depression). The comparison of the groups (Table 1) again showed significant age differences. The pure dementia cases were about three years older than those of the other groups (ANOVA df 3, F = 20.8 ***). Gender was evenly distributed. 218 of 323 dementia cases (67.5%) had DSD. Only 46 of 264 delirium cases (17.4%) had no dementia symptomatology. The four groups did not differ regarding the extent of infarcts or WMH.

The comparison of predisposing factors for delirium in patients with dementia alone and DSD reveals some significant differences (Table 2). The DSD patients were younger than those with pure dementia (ANOVA df 1 F = 5.97 *) and were more severely demented as assessed by CDR (ANOVA df 1 F = 44.9 ***). Furthermore, their nutritional status is lower (ANOVA df = 1 F = 18.3 ***). No significant differences are found for disabilities (visual impairment, hearing loss, or restricted mobility) or CT findings. Logistic regression, including the variables number of ICD-10 diagnoses, CIRS-13, and MNA score, is performed controlling for age, gender, and dementia stage. Only the dementia stage and younger age significantly contributed to the differentiation between pure dementia and DSD. CIRS-13 data measuring MM revealed no differences between neurocognitive disorders (Table 1) nor in DSD patients with probably different underlying brain damage (Table 3).

In the third step, only the 218 cases classified as DSD were considered with regard to the predisposing factors. They were subdivided into three groups according to the CT scans: no changes or atrophy only (=probably neurodegenerative dementia), with infarcts (=probably vascular dementia), and those with WMH (=probably cerebral microangiopathy). As shown in Table 3, those cases without signs of a vascular etiology were significantly younger (ANOVA df 2 F 4.0 *). Those with infarcts had a trend to more severe MM as estimated by the CIRS-13 (n.s.) and a higher rate of visual impairment (Chi2 8.99 *) and restricted mobility (n.s). The cases with WMH suffered from less severe dementia as estimated by CDR (ANOVA F = 3.25 *)

In Table 4, the potentially precipitating factors in demented patients with and without delirium were presented. A lot of laboratory parameters indicating, i.e., diabetes mellitus (Hba1c) anemia, etc., showed slightly elevated odds ratios, but the corresponding confidence intervals started at values clearly below one. Thus, no elevated risk can be concluded. Only a current infection can be estimated as a trigger of DSD. The number of administered psychotropic drugs and of other medications at admission had no effect on the development of DSD.

Considering the potentially underlying morphological brain changes of DSD (Table 5), the comparison revealed no differences in the tested laboratory parameters. The number of applied drugs at admission showed inconsistencies that are hard to interpret. While the number of administered psychotropic drugs was significantly higher in DSD patients without vascular signs (ANOVA F = 6.10 **), the number of other medications was highest in DSD patients showing WMH (ANOVA F = 6.10 **).

To evaluate the outcome, the CCI score, the length of stay, and the general status at discharge as assessed by CGI were evaluated. The calculation revealed no differences between the DSD groups besides a higher CCI score in DSD patients with brain infarction (ANOVA F = 3.88 *).

## 4. Discussion

Dementia and delirium have been conceptualized as distinct and mutually exclusive conditions according to common guidelines [10,11]. Both diagnostic guidelines state that dementia should not be diagnosed in the face of delirium. Furthermore, delirium should not be diagnosed when symptoms can be “better accounted for by a pre-existing, established, or evolving dementia”. Distinguishing the two diagnoses in the clinical setting can be difficult, however, even for experienced clinicians [6]. 

Delirium can be thought of as “acute brain failure”, a multifactorial syndrome analogous to acute heart failure. The onset of delirium involves a complex interaction between the patient’s baseline vulnerability (predisposing factors) and precipitating factors or noxious insults. The precipitating factors vary due to the investigated sample of patients [13,14]. Most studies concerning DSD were conducted in perioperative or ICU patients [35]. Only a few studies reported data collected from general medicine [21,27,36,37]. In this investigation, based on the GePsy B databank, all patients admitted after stroke, surgical operation, or suffering from probable Parkinson’s or Lewy body disease or alcohol- or medication-withdrawal were excluded. Thus, the investigated sample differs in important aspects from most of the previous studies concerning DSD. 

Moreover, the comparison of the published studies is hindered by partly different scales to assess delirium, i.e., CAM-S [26] or DRS-98 [28]. However, these measures have a high degree of correlation in hospitalized patients [37,38], also with the DSM-IV or ICD-10 criteria [37]. Previous studies suggested that DSD patients may have a more severe course of delirium [36]. However, in our sample, delirium severity does not differ between delirium alone and DSD, nor between the DSD groups subdivided according to CT scans. The CDR scores even show less severe dementia stages in DSD patients with WMH. These results suggest that the psychopathology of DSD may be influenced by the probably underlying type of brain disorder.

Since many predisposing factors of delirium are known [13,14], it seemed obvious that physical MM may be an essential contributor to delirium and DSD. However, our CIRS-13 data measuring MM revealed no differences between delirium and DSD. The number of ICD-10 diagnoses used as an indicator for MM showed no significant differences. In summary, MM indices do not provide a simple way to predict the development of DSD.

A systematic review and meta-analysis supported the existence of a relationship between frailty and delirium, although there was notable methodological heterogeneity between studies [39]. In our study corresponding, some parameters, such as malnutrition and physical disability, were assessed. The frequency of visual impairment, hearing loss, and restricted mobility showed no significant differences between dementia and DSD. These results agree with those in very old patients (>80 years [37]. However, impaired sight and restricted mobility seemed to be more frequent in DSD patients with brain infarction.

Many risk factors for delirium are known. However, there is a disparity concerning the investigated sample of patients (i.e., ICU, after surgical operation) [13]. The data of our sample of hospitalized medical patients suggest that infections and malnutrition may influence the development of DSD, but that MM does not. The influence of WMH ([18], this study) has to be confirmed by further studies.

While previous studies revealed that DSD patients may have poorer outcomes, including increased length of hospitalization, poorer functional status, and higher mortality [36,40], our data showed no such poorer outcome, not even in the CCI or CGI at discharge.

## 5. Limitations

There are several limitations to this study. First, our study was based on the retrospective analysis of clinical data of patients consecutively admitted to only one hospital.

Moreover, there may be selection bias. About 20% of our patients were transferred from medical or surgical wards or the ICU of our hospital. About 30% were primarily admitted to the central emergency room and transferred to our ward after excluding life-threatening diseases. This is likely to have reduced the number of patients with delirium, in whose case medical comorbidities could be identified.

Second, there is a possible selection bias as many cases had to be excluded because in compliance and agitation made CT scans impossible. It can be assumed that these cases were more mentally impaired. 

As no established biomarkers for delirium exist, DSD is diagnosed clinically. The differentiation of the DSD patients according to CT scans into atrophy only, with infarction and with WMH, does not allow a well-grounded pathophysiological categorization. However, all patients showing in CT scans brain damage (such as a tumor or subdural hematoma, etc.) and all patients with a history of extrapyramidal symptoms prior to dementia were excluded from analysis in order to focus on typical cerebral changes associated with dementia since the frequency of delirium and its relation to dementia in Parkinson’s disease [41] or Lewy body disease remains unclear [42]. CSF levels of β-42 amyloid, total tau, and phospho-tau protein and apolipoprotein E were not routinely measured, such as in some other studies [43], but studies of these AD biomarkers in delirium revealed mixed results [44].

In summary, despite the limitations, we think that the chosen categorization of DSD according to morphological findings in CT scans might help to elucidate the underlying brain disorders of DSD.

## 6. Conclusions

Our results suggest that MM and the underlying brain disorder are not predisposing factors of DSD in patients living with dementia. Younger age and higher dementia stage affect the development of DSD. The only potentially precipitating factor for delirium in demented patients found in this study is current infections.

## Figures and Tables

**Figure 1 geriatrics-08-00064-f001:**
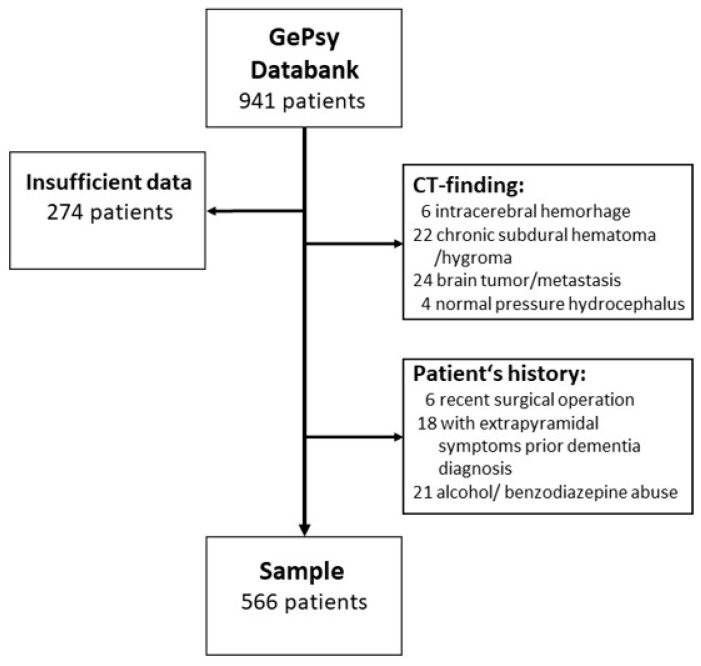
Flow diagram showing the selection of study patients.

**Table 1 geriatrics-08-00064-t001:** Sample.

	Other Diagnoses	Dementia Alone	Delirium Alone	Delirium Super- Imposed Dementia	Statistics ANOVA
n	197	105	46	218	
Age (range 66–101)	76.6 ± 7.5	80.1 ± 7.2	77.5 ± 8.0	76.6 ± 7.5	F = 20.8 df 3 *p* < 0.001
Gender [male]	31.5%	30.5%	34.8%	33.9%	Chi^2^ 0.61 df 3 *p* = 0.895
CDR (max. 3)	0.1 ± 0.2	1.6 ± 0.6	0.2 ± 0.3	2.2 ± 0.8	F = 508.8 df 3 *p* < 0.001
CAM-S (max. 7)	2.2 ± 1.6	3.4 ± 1.3	5.5 ± 1.4	5.9 ± 1.7	F = 211.8 df 3 *p* < 0.001
DRS-98 (max. 16)	5.2 ± 2.4	7.8 ± 2.0	11.1 ± 2.4	11.4 ± 2.5	F = 250.7 df 3 *p* < 0.001
CIRS 13 (Max. 52)	12.7 ± 3.5	13.3 ± 3.8	13.4 ± 4.0	12.9 ± 3.8	F = 36.9 df 3 *p* = 0.890
cCT findings					
Infarcts	18 (22.0%)	18 (17.1%)	7 (15.2%)	39 (17.9%)	Chi^2^ 7.21 df 3 *p* = 0.066
WMH	104 (52.8%)	68 (64,8%)	29 (63.0%)	123 (56.4%)	Chi^2^ 4.71 df 3 *p* = 0.194

**Table 2 geriatrics-08-00064-t002:** Comparison of predisposing factors for delirium in patients with dementia alone and DSD.

	Dementia Alone	Delirium Superimposed Dementia	Statistics
n	105	218	
Age (range 66–101)	80.1 ± 7.2	76.6 ± 7.5	ANOVA F = 5.97 df 1 *p* = 0.015
Gender [male]	30.2	33.9%	Odds ratio 0.85 (95% CI 0.51…1.41)
CDR (max. 3)	1.6 ± 0.6	2.2 ± 0.8	ANOVA F = 44.9 df = 1 *p* < 0.001
Multimorbidity			
Number of ICD-10 diagnoses	10.5 ± 3.3	11.1 ± 3.7	ANOVA F= 1.93 df = 1 *p* = 0.165
CIRS-13 (max. 52)	13.3 ± 3.8	12.9 ± 3.8	ANOVA F = 0.81 df = 1 *p* = 0.370
MNA- SF (max. 14)	7.3 ± 1.6	6.5 ± 1.7	ANOVA F = 18.3 df = 1 *p* < 0.001
Disability			
Visual impairment	8 (7.6%)	19 (8.7%)	Odds ratio 1.16 (95% CI 0.49…2.74)
Hearing loss	16 (15.2%)	46 (21.1%)	Odds ratio 1.49 (95% CI 0.80…2.78)
Restricted mobility	30 (28.6%)	58 (26.6%)	Odds ratio 0.91 (95% CI 0.54…1.52)
cCT findings			
Infarcts	18 (17.1%)	39 (17.9%)	Odds ratio 1.05 (95% CI 0.57…1.95)
WMH	68 (64.8%)	123 (56.4%)	Odds ratio 0.73 (95% CI 0.46…1.19)

**Table 3 geriatrics-08-00064-t003:** Predisposing factors for delirium in DSD patients with different underlying brain damage.

cCT-Finding	Atrophy Only	Infarction	WMH	Statistics
n	56	39	123	
Age [years]	79.9 ± 6.6	82.8 ± 7.4	83.0 ± 7.0	F = 4.00 df 2 *p* = 0.020
Gender [% male]	41.1%	38.5%	29.3%	Chi^2^ 2.82 df 2 *p* = 0.244
CDR (max. 3)	2.3 ± 0.7	2.3 ± 0.7	2.1 ± 0.8	F = 3.25 df 2 *p* = 0.040
CAM-S (max. 7)	6.0 ± 1.8	5.9 ± 1.5	5.9 ± 1.7	F = 0.04 df 2 *p*= 0.961
DRS-98 (max. 16)	11.4 ± 2.7	12.1 ± 2.5	11.2 ± 2.3	F = 1.96 df 2 *p* = 0.143
Multimorbidity				
Number of ICD-10 diagnoses	10.7 ± 4.0	11.6 ± 3.3	11.2 ± 3.7	F = 0.83 df 2 *p* = 0.438
CIRS-13 (max. 52)	12.8 ± 3.7	14.1 ± 3.5	12.5 ± 4.0	F = 2,78 df 2 *p* = 0.064
MNA-SF (max. 14)	6.4 ± 1.5	6.6 ± 1.7	6.5 ± 1.7	F = 0.19 df 2 *p* = 0.829
Disability				
Visual impairment	2 (3.6%)	8 (20.5%)	9 (7.3%)	Chi^2^ 8.99 df 2 *p* = 0.011
Hearing loss	8 (14.3%)	9 (23.1%)	29 (23.6%)	Chi^2^ 2.11 df 2 *p* = 0.349
Restricted mobility	19 (33.9%)	14 (35.9%)	25 (20.3%)	Chi^2^ 5.75 df 2 *p* = 0.057

**Table 4 geriatrics-08-00064-t004:** Potentially precipitating factors in demented patients with and without delirium.

	Dementia Alone	Delirium Superimposed Dementia	Statistics
Elevated Hba_1c_ (>6.5%)	48 (45.7%)	73 (33.5%)	Odds ratio 0.60 (95% CI 0.37 …0.96)
Leukocytosis (>9000/µL)	15 (14.3%)	37 (17.0%)	Odds ratio 1,23 (95% CI 0.64 …2.35)
Anemia (Hb females < 12 g/L, males < 14 g/L)	60 (57.1%)	121 (55.5%)	Odds ratio 0.94 (95% CI 0.59 …1.50)
Hematocrit (%)	15 (14.3%)	41 (18.8%)	Odds ratio 1.39 (95% CI 0.73 …2.65)
Hyponatremia (<135 mmol/L)	7 (6.7%)	17 (7.8%)	Odds ratio 1.18 (95% CI 0.48 …2.95)
Elevated creatinine (>90 mg/L)	47 (44.8%)	92 (42.2%)	Odds ratio 0.90 (95% CI 0.56 …1.44)
TSH (<0.25 µU/mL)	9 (8.6%)	20 (9.2%)	Odds ratio 1.08 (95% CI 0.47 …2.46)
Current Infection	29 (27.6%)	86 (39.4%)	Odds ratio 1.71 (95% CI 1.03 …2.83)
Dehydration	22 (21.0%)	57 (26.1%)	Odds ratio 1.33 (95% CI 0.76 …2.34)
Medication			
Psychotropic drugs	1.7 ± 1.1	1.9 ± 1.0	ANOVA F 2.08 df 2 *p* = 0.150
Other medication	4.8 ± 2.9	4.5 ± 2.6	ANOVA F 0.99 df 2 *p* = 0.320

**Table 5 geriatrics-08-00064-t005:** Potentially precipitating factors in DSD-patients.

cCT Finding	No Or Atrophy	Infarction	WMH	Statistics
Elevated Hba_1c_ (>6.5%)	19 (33.9%)	10 (25.6%)	44 (35.8%)	Chi^2^ 1.37 df 2 *p* = 0.504
Leukocytosis (>9000/µL)	6 (10.7%)	6 (15.4%)	25 (20.3%)	Chi^2^ 2.61 df 2 *p* = 0.272
Anemia (Hb females < 12 g/ L, males < 14 g/L)	36 (64.3%)	23 (59.0%)	62 (50.4%)	Chi^2^ 3.23 df 2 *p* = 0.199
Hematocrit (%)	7 (12.5%)	8 (20.5%)	26 (21.1%)	Chi^2^ 1.97 df 2 *p* = 0.373
Hyponatremia (>135 mmol/L)	3 (5.4%)	4 (10.3%)	10 (8.1%)	Chi^2^ 0.81 df 2 *p* = 0.667
Elevated creatinine (>90 mg/L)	21 (37.5%)	14 (35.9%)	57 (46.3%)	Chi^2^ 2.00 df 2 *p* = 0.365
TSH (<0.25 µU/mL)	5 (8.9%)	1 (2.6%)	14 (11.4%)	Chi^2^ 2.77 df 2 *p* = 0.250
Current infection	18 (32.1%)	18 (46.2%)	50 (40.7%)	Chi^2^ 2.00 df 2 *p* = 0.357
Dehydration	13 (23.2%)	9 (15.8%)	35 (28.5%)	Chi^2^ 0.78 df 2 *p* = 0.677
Restricted mobility	19 (33.9%)	14 (35.9%)	25 (20.3%)	Chi^2^ 5.75 df 2 *p* = 0.057
Medication				
Psychotropic drugs (range 0–4)	2.3 ± 1.0	1.7 ± 1.0	1.8 ± 1.0	ANOVA F = 6.10 df 2 *p* = 0.003
Other medication (range 0–15)	3.5 ± 2.1	4.5 ± 2.8	5.0 ± 2.7	ANOVA F = 6.10 df 2 *p* = 0.003

## Data Availability

No new data were created or analyzed in this study. The data presented in this study are available on request from the corresponding author.

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
