# Peer review of "Contribution of Different Brain Disorders and Multimorbidity to Delirium Superimposed Dementia (DSD)"

_geriatrics, 2023, doi:10.3390/geriatrics8030064_

Round 1
Reviewer 1 Report
Mant thanks for asking me to review this paper on structural brain disorders, MM and delirium
Abstract. Personally I don't like narrative abstracts but I realise that this is the journals choice. However, with fresh eyes, could you consider rewording it, as it does not read particularly clearly. Especially in terms of exactly what you are trying to look, how you did it and what you found
Background. Generally well written. Please rewrite lines 54 and 55, there are errors. The final paragraph, first line, please rewrite as 'This study will assess to hypotheses.'
Methods: It is unusual to quote a reference directly [24]. Line 69, replace impossibility to perform with 'difficulties in performing'
Figure 1. Flow diagram...
Results
First line, a three step (not steps)
Conclusions:
Are noT predisposing factors.... In patients living with demrntia
Modest improvement requied
Reviewer 2 Report
The authors asked if they could distinguish dementia from delirium superimposed on dementia on the basis of two patient characteristics: multimorbidity and brain imaging evidence of cerebral atrophy and/or white matter hyperintensities.
The topic is relevant to medial practice, geriatric psychiatry, and CL psychiatry, in that many elderly patients, whether at home, hospitalized, or in nursing homes, have dementia. Recognizing that some have delirium may be a trigger for medical interventions to resolve the precipitants of the delirium, such as infection or metabolic disarray.
A main limitation of the paper, as I understand it, is that the patients are all from a neuropsychiatry ward, rather than from general medical and surgical services. This is likely to have reduced the number of patients with delirium in whose case medical comorbidities could be identified that may have helped distinguish superimposed delirium from dementia alone. Is the sample restricted to pts admitted to a neuropsychiatric unit? Rather than acutely ill pts admitted to a general hospital? This would likely affect the results.
Table 4: What is “exsiccosis?” I’ve never encountered this term before. In English the following spelling is correct: Leukocytosis.
Ms page 6: “psychotropic” not “psychotrophic; ” …inconsistencies that are “hard” to interpret…, not “hardly” to interpret; …the number of other “medications” was highest, not “medication”
Mostly ok--a few oddities and errors noted
